# RobustSTL and Machine-Learning Hybrid to Improve Time Series Prediction of Base Station Traffic

**Chih-Hsueh Lin and Ulin Nuha \***

Department of Electronic Engineering, National Kaohsiung University of Science and Technology, Kaohsiung 80778, Taiwan; cslin@nkust.edu.tw
**\*** Correspondence: i110152118@nkust.edu.tw

**Abstract:** Green networking is currently becoming an urgent compulsion applied for cellular network architecture. One of the treatments that can be undertaken to fulfill such an objective is a traffic-aware scheme of a base station. This scheme can control the power consumption of the cellular network based on the number of demands. Then, it requires an understanding of estimated traffic in future demands. Various studies have undertaken experiments to obtain a network traffic prediction with good accuracy. However, dynamic patterns, burstiness, and various noises hamper the prediction model from learning the data traffic comprehensively. Furthermore, this paper proposes a prediction model using deep learning of one-dimensional deep convolutional neural network (1DCNN) and gated recurrent unit (GRU). Initially, this study decomposes the network traffic data by RobustSTL, instead of standard STL, to obtain the trend, seasonal, and residual components. Then, these components are fed into the 1DCNN-GRU as input data. Through the decomposition method using RobustSTL, the hybrid model of 1DCNN-GRU can completely capture the pattern and relationship of the traffic data. Based on the experimental results, the proposed model overall outperforms the counterpart models in MAPE, RMSE, and MAE metrics. The predicted data of the proposed model can follow the patterns of actual network traffic data.

**Keywords:** network traffic prediction; base station; green networking; RobustSTL; machine learning





## 1. Introduction

The implementation of green networking architecture nowadays attracts attention. That is, the base station architecture must be power saving. About 70% of the power in the cellular network infrastructure is burned by base station units [1]. Hence, the first undertaking is to predict future network traffic of the base station. A network traffic prediction can bring significant information for understanding traffic patterns [2]. Through such prediction, the base station may actively control its power demands by reducing the bandwidth capacity during low traffic to lower the energy consumption of the base station. However, nonlinearity and intricacy in traffic data are the main issues in network prediction [3].

Generally, the effort of network traffic prediction is categorized into two schemes, i.e., model driven (parametric) and data driven [4]. The first scheme works based on the practicality of the theoretical assumptions, such as autoregressive integrated moving average (ARIMA) model, and the second scheme deals with machine learning by interpreting and learning the data, such as artificial neural network (ANN). Various models have been applied in network traffic prediction studies. Zhang et al. [5] presented an improved long short-term memory (LSTM) with wavelet transform to decompose the original internet network traffic. This model can successfully reduce the prediction error in the network traffic prediction problem. In Ref. [6], a method integrated with fuzzy clustering and the weight exponential to improve LSTM and adaptive neuro-fuzzy inference system (ANFIS) in series models was proposed. From the results, this proposed method can increase the

prediction accuracy rate by enhancing the reliability of the preprocessing stages. Wavelet neural network (WNN) with seeker optimization algorithm (SOA) based on the dynamic adaptive search step was proposed to optimize the prediction accuracy by overcoming poor local search and adaptive adjustment ability of traditional SOA [2]. This model can catch the trend of the traffic data signal and has the validity of the prediction accuracy, but this model may not be robust for long-term prediction. Zheng et al. proposed a method for 4G network base station prediction by combining the $v$ support vector regression ($v$SVR) algorithm with the optimization of a symbiotic organisms search (SOS) [7]. The obtained optimal prediction result takes a lot of experiments in the input optimization.

However, the existing research in network traffic data faces challenges affecting the prediction accuracy. The main challenges are complicated characteristics and dynamics patterns. Zhang et al. [5] designed a model of network traffic prediction by utilizing wavelet transform to decompose the original data into multiple components with different frequencies as the input of the model. These components can bring significant trends of time granularity to learn the changing rules of the traffic. In Ref. [8], a seasonal and trend decomposition using Loess (STL) was performed to address noises of the network traffic. STL decomposes the network traffic into seasonal, trend-cycle, and remainder components. The result of these components is then utilized as input for the GRU model. Another study combined LSTM and Gaussian process regression (GPR) to accurately predict single-cell cellular [9]. First, it extracts the periodic components of data traffic using Fourier transform fed into the LSTM cell, while GPR predicts the residual random components. This proposed method can successfully increase prediction accuracy compared to the traditional method.

Although some models have reached well the prediction of the network traffic with dynamic traffic patterns, they still need to improve in identifying abrupt changes, traffic burstiness, and outliers in future demands. Furthermore, this study proposes deep learning of base station traffic prediction with the reliability of traffic burstiness and outliers. To obtain the comprehensive pattern of traffic load from the base station, this study first decomposes the original traffic data using RobustSTL method, instead of the standard STL. RobustSTL not only decomposes seasonal, trend, and remainder or residual components but can also further decompose the remainder component into a spike and white noise [10]. The resulted components are synchronously fed into a hybrid model of one-dimensional convolutional neural network (1DCNN) and gated recurrent units (GRU). Through this hybrid model, 1DCNN can catch and extract features of the components [11], and GRU can capture the rules and relationships among the components to improve model performance [12]. Although there is a study in the load forecasting field proposing 1DCNN-GRU [13], it does not utilize decomposition. Furthermore, we enhance their proposed model that not only proposes the hybrid model of 1DCNN and GRU but also utilizes the decomposition of RobustSTL in the beginning stage.

Through the above analysis, this paper focuses on designing a deep-learning method for base station traffic prediction by combining RobustSTL and 1DCNN-GRU. The primary contributions of this paper can be summarized as follows:

1. We propose a single traffic load prediction model of the base station based on RobustSTL and 1DCNN-GRU. This method can extract dynamics patterns of traffic data and more accurately predict the base station traffic;
2. The main contribution is the hybrid model of the decomposition of time series data using RobustSTL technique, instead of the standard STL, with 1DCNN-GRU;
3. The proposed model can give the reference for sleeping control operation in the base station by estimating the future internet traffic.

The remaining sections of this paper are organized as follows. In Section 2, this paper gives a background on our materials and the methods used in this study. Section 3 presents the experimental results of base station traffic prediction and discusses these results. In Section 4, we put forward the conclusions of this study and highlight the directions of future research.

## 2. Materials and Methods

### 2.1. RobustSTL and 1DCNN-GRU

As we mentioned above, we use the combination model of RobustSTL and 1DCNN-GRU to predict base station traffic. Figure 1 shows that a raw input is decomposed by RobustSTL to reveal the underlying insights of base station traffic. After obtaining three decomposed components, i.e., the trend, seasonality, and residual components, these are simultaneously fed into 1DCNN. 1DCNN can help a better understanding of traffic patterns and spatial features. The 1DCNN architecture here consists of convolution, pooling, and fully connected layers. The number of inputs in the 1DCNN model from one raw input is 3 (three), obtained from the decomposition result. Then, these 1DCNN inputs are fed to the convolutional layer, which has 32 filters. The outputs of the convolutional layer are carried to the pooling layer to minimize the dimension. The last layer in the 1DCNN is the fully connected layer where the input is from the pooling layer.

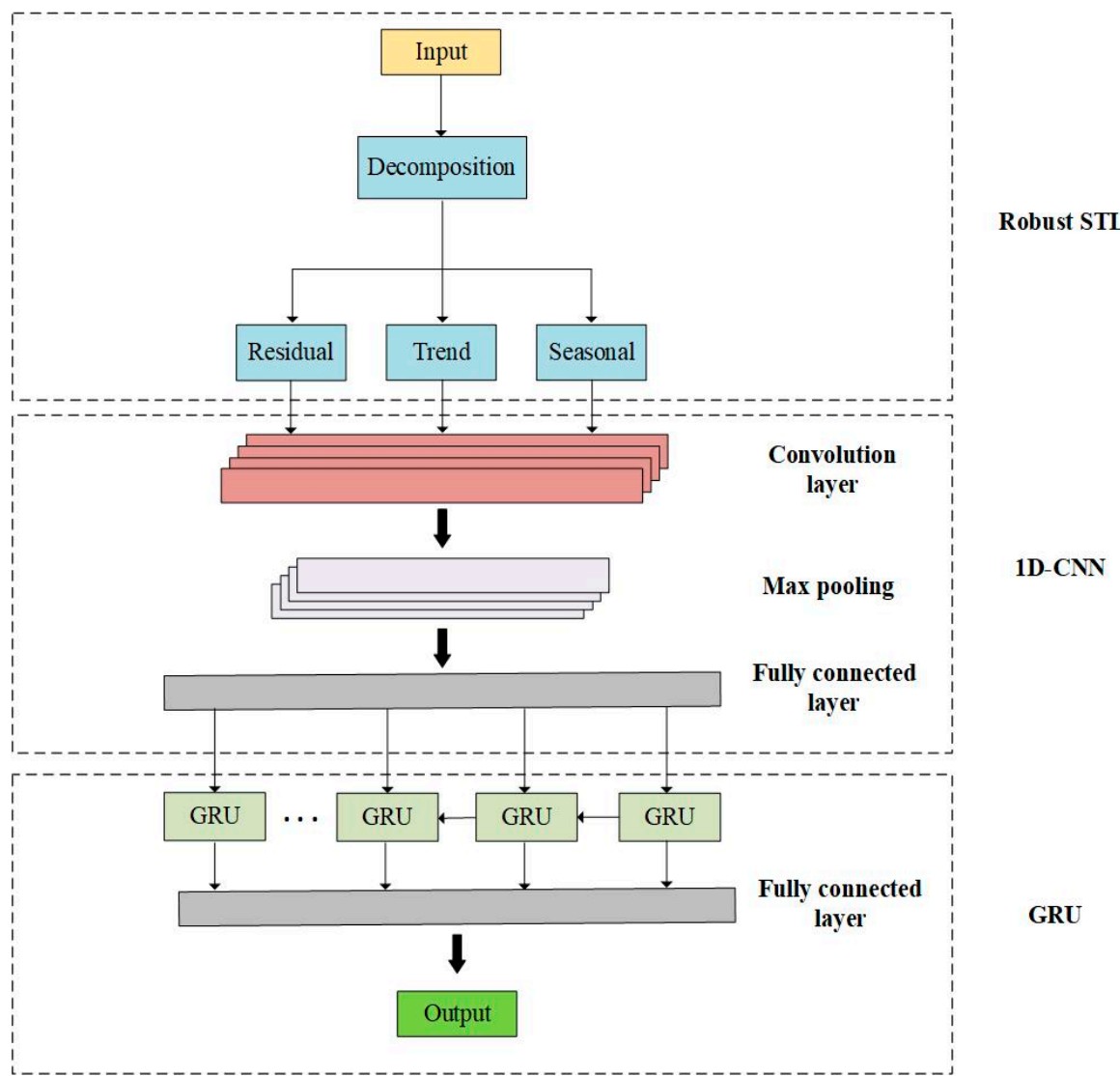

**Figure 1.** Robust STL and 1DCNN-GRU architecture.

The outputs from the 1DCNN block are put into the GRU block. Through the GRU operation, the traffic characteristic and rules can be learned to improve the accuracy of the prediction result. The number of GRU cells is the same as the number of 1DCNN

outputs. Then, the outputs of GRU are flattened with the fully connected layer to obtain the final output.

### 2.2. Standard STL

STL is a statistical method for decomposing a time series data into three elements, i.e., seasonality, trend, and residual components. Trend is a systematic pattern that changes over time, does not repeatedly happen, and indicates the general tendency of the data to increase or decrease during a period. Seasonality is a component that repeatedly changes over time and represents data fluctuation. Meanwhile, remainder is a non-systematic component besides trend and seasonality components within the data. Suppose $y_t$ is time series data, STL decomposes the time series data into seasonality $s_t$, trend $\tau_t$, and residual $r_t$ [14] as follows:

$$y_t = s_t + \tau_t + r_t, \text{ for } t = 1 \text{ to } T. \tag{1}$$

The standard STL decomposition consists of two loops, i.e., inner loop and outer loop [15]. The inner loop upgrades seasonal and trend components in each of the passes. Then, the residual component is calculated. The outer loop will be adjusted by the Loess, when an anomaly is detected.

### 2.3. RobustSTL

In standard STL, we assume that all components contained in the time series data besides trend and seasonality are residual components. In contrast, RobustSTL proposed by Wen et al. [10] and adopted in this study claims that residual components can be further extracted into two terms, i.e., spike and noise. Moreover, RobustSTL can accurately and precisely decompose the time series data despite containing a long seasonality and high noise [10]. This method can handle fractional and shifted seasonality components over a period.

Data collected from the management unit frequently contain various noises. To obtain the exact trend and seasonality components, such noises initially need to be removed. Here, RobustSTL proposes a bilateral filtering to remove various noises. Then, RobustSTL utilizes the least absolute deviations (LAD) with L1 norm regularizations to extract the trend component and non-local seasonal filtering to obtain the seasonality component. Given a data of time series, Algorithm 1 shows the decomposition summary of the RobustSTL method.

---

**Algorithm 1:** RobustSTL decomposition summary.

---

**Input**: $y_t$, parameter configuration
**Output**: $s_t, \tau_t, r_t$
1: Denoise the network traffic data $y_t$ by bilateral filtering to obtain denoised data $y'_t$
2: Obtain the relative trend $\overline{\tau}^r_t$ and apply this equation $y''_t = y'_t - \overline{\tau}^r_t$ to denoised data
3: Perform the seasonality extraction to $y''_t$ using non-local seasonal filtering to obtain
　　$\bar{s}_t$ value
4: Obtain trend, seasonality, and residual components
　　$\tau_t = \overline{\tau}^r_t + \tau_1, s_t = \bar{s}_t - \tau_1, r_t = y_t - s_t - \tau_t$
5: Repeat steps 1–4 to obtain more accurate estimation

---

### 2.4. 1DCNN

In the proposed model architecture, we introduce one-dimensional convolutional neural network (1DCNN) for the component of the base station traffic prediction model. 1DCNN can extract the morphological features of the traffic data to enhance the understanding of traffic patterns [11]. 1DCNN architecture generally consists of convolutional layer, pooling layer, and fully connected layer.

#### 2.4.1. Convolutional Layer

This layer in 1DCNN overcomes the regular neural network by a faster convergence. This layer contains a set of time series maps, kernels, filters, strides, and neurons. This

layer connects each neuron to the neighbor neurons. A convolutional operation calculates the dot product between corresponding convolution filters and the input time series maps. Through the kernels, this can learn a characteristic of the input maps. The time series input is first fed to the input layer. Then, the output of the convolution process is calculated as follows:

$$\overline{y}_m^l = b_m^l + \sum_{i \in L_m} conv1D(\overline{w}_{im}^{l-1}, \hat{y}_i^{l-1}), \tag{2}$$

where $\overline{y}_m^l$ and $b_m^l$ are the output and the bias of the $m^{th}$ neuron at layer $l$. $\overline{w}_{im}^{l-1}$ is kernel weight of the $i^{th}$ neuron at layer $l-1$, and $\hat{y}_i^{l-1}$ is the input to the $m^{th}$ neuron at layer $l$ from the $i^{th}$ neuron at layer $l-1$. $L_m$ denotes a selection of input maps. Then, $conv1D$ refers to the convolution operator.

### 2.4.2. Max Pooling Layer

Pooling operation in the 1DCNN model refers to the resampling process, i.e., a transformation of multiple cells into one cell. This operation has advantages such as minimizing the computational cost while retaining the significant information. Besides, this operation can also avoid an overfitting model [16]. In this study, we select a max pooling model, which takes a maximum value of an array.

Figure 2 shows an example max pooling operation of time series data. Initially, the total element of the time series map is (15), and the max pooling with a stride of three (3) divides them into five (5) groups denoted by various colors. Then, we can obtain the smaller size of the time series map maintaining the discriminant information through the following equation:

$$p = \max(\overline{y}_R), \tag{3}$$

where $p$ is the output of max pooling operation, and $\overline{y}_R$ is the elements of corresponding pool area $R$.

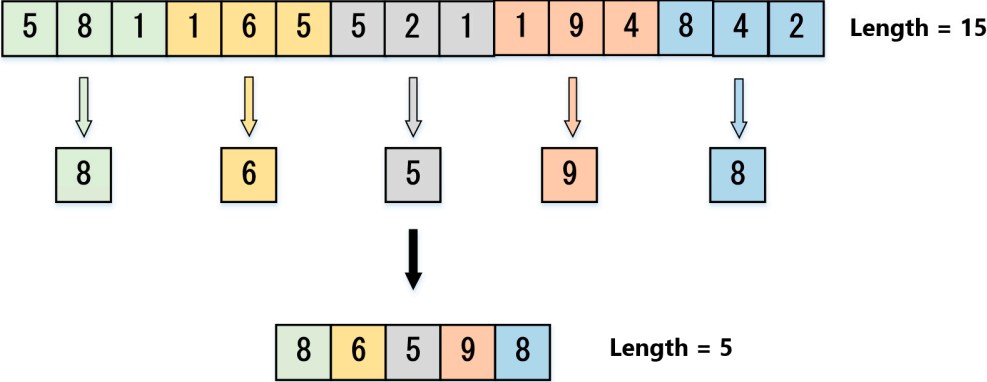

**Figure 2.** Max pooling operation [16].

### 2.4.3. Fully Connected Layer

The fully connected layer receives the outputs of the pooling layer. First, they are typically flattened to be converted into a one-dimensional array, then connected to a layer where each input is linked to outputs with a weight. This layer has a similar structure to regular ANN, and a neuron in this layer can be calculated as follows:

$$u_k = \sum_{i=1}^m \omega_{i,k} p_i + b_k, \tag{4}$$

where $u_k$ is an output value at the $k^{th}$ neuron, and $\omega_{i,k}$ is a weight value of $i^{th}$ input at the $k^{th}$ neuron. Meanwhile, $p_i$ is the $i^{th}$ input value, and $m$ is the number of inputs. Then, $b_k$ is the bias value at the $k^{th}$ neuron. After we obtain the output of the fully connected layer, this layer is followed by an activation function, such as rectified linear unit (ReLU) or linear activation function.

The GRU architecture is a simpler model compared to the LSTM architecture because the GRU is without memory cells. However, it has a low computational cost but the same performance result as LSTM performance [17]. GRU architecture consists of two cells, i.e., update gate and reset gate. The update gate determines the past information of time series to be remembered or forgotten for the current prediction need. Then, the reset gate determines the amount of information to be remembered. Figure 3 depicts the architecture of GRU.

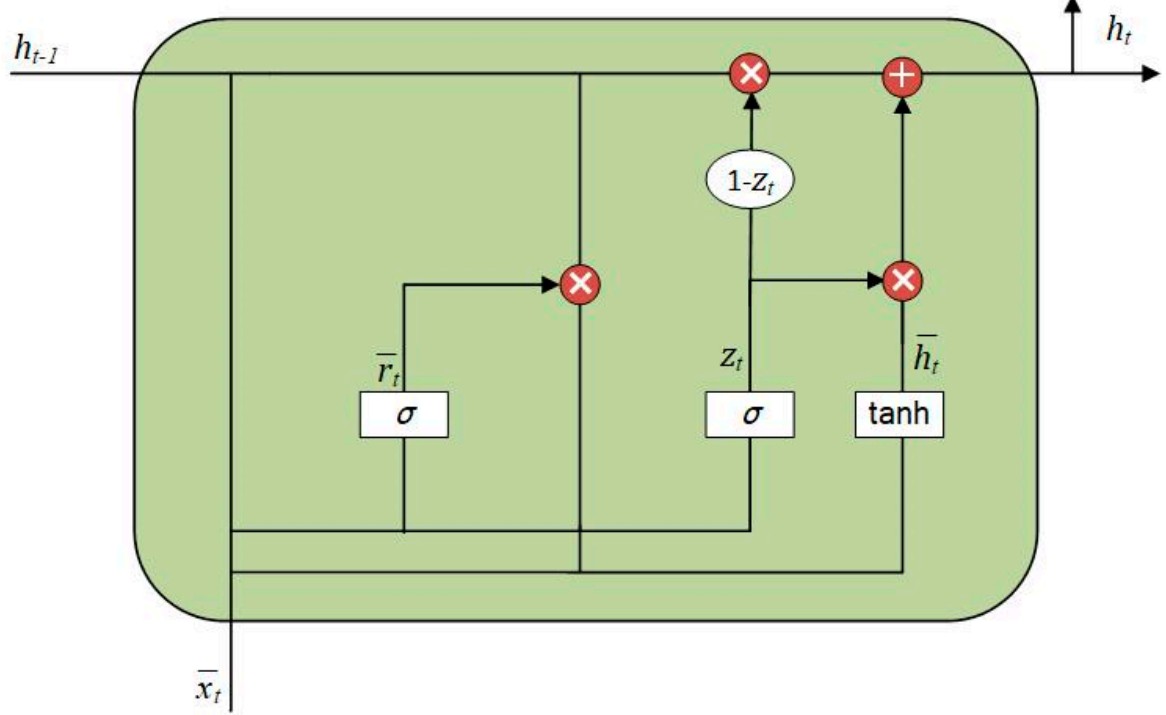

**Figure 3.** GRU architecture [18].

Initially, the current input $\overline{x}_t$ and the previous output $h_{t-1}$ are concatenated to be the input for the update gate and the reset gate. Furthermore, each value of all cells or components in GRU can be obtained as follows:

$$z_t = \sigma(W_z \cdot [h_{t-1}, \overline{x}_t] + b_z), \tag{5}$$

$$\overline{r}_t = \sigma(W_{\overline{r}} \cdot [h_{t-1}, \overline{x}_t] + b_{\overline{r}}), \tag{6}$$

$$\overline{h}_t = \tanh(W_{\overline{h}} \cdot [\overline{r}_t \cdot h_{t-1}, \overline{x}_t] + b_{\overline{h}}), \tag{7}$$

where $z_t$, $\overline{r}_t$, and $\overline{h}_t$ are the outputs of the update gate, the reset gate, and the candidate hidden layer. $\sigma$ and $\tan$ h are the activation functions. Then, $W_z$, $W_{\overline{r}}$, $W_{\overline{h}}$ and $b_z$, $b_{\overline{r}}$, $b_{\overline{h}}$ are, respectively, weight matrix and bias vectors of the update gate, the reset gate, and the candidate hidden layer. Finally, we can obtain the output $h_t$ of the $t^{th}$ GRU as follows:

$$h_t = (1 - z_t) \times h_{t-1} + z_t \times \overline{h}_t. \tag{8}$$

## 3. Numerical Results and Performance Analysis

*3.1. Preparation Stage*

The dataset used in the experiment stage is received from Italy cellular network activity from Kaggle [19]. In this dataset, there is base station internet traffic from lots of locations. Each location is represented in the cell id in the dataset as instances. Here, we take internet

traffic in three locations. They are cell 3350, cell 5060, and cell 5864. Figure 4 shows a closer view of the internet network traffic over an hourly period in the selected base station.

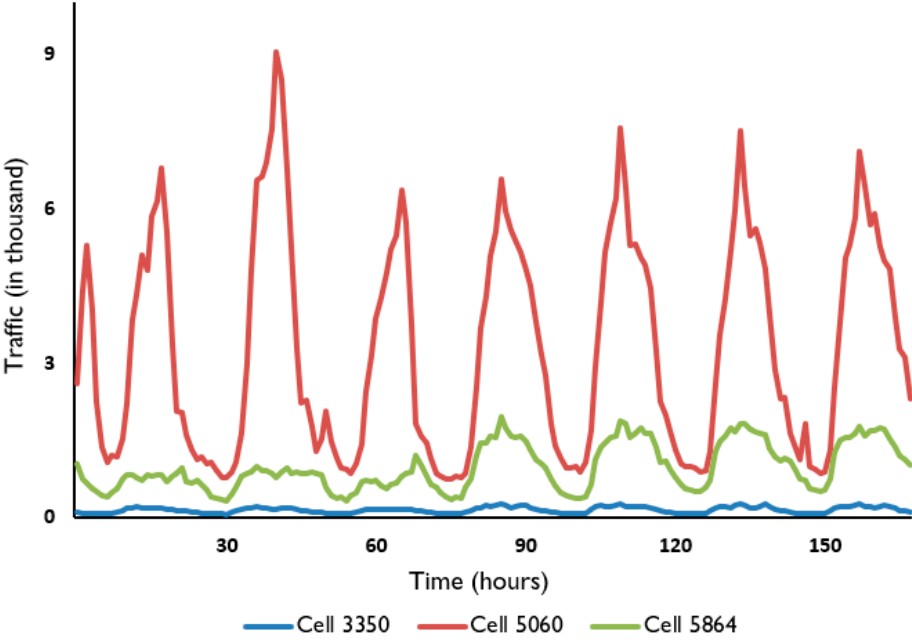

**Figure 4.** Internet traffic of base station over a period.

Before we go into the training process, the training dataset is processed with a normalization to lead to stability, especially in the datasets containing various noises [20]. The normalization formula of Min–Max Scaler is presented in the equation below:

$$\widetilde{x} = \frac{x - x_{\min}}{x_{\max} - x_{\min}}, \tag{9}$$

where $x$ is the original data, while $x_{\min}$ and $x_{\max}$ are the minimum value and the maximum value of the dataset. Then, $\widetilde{x}$ is the normalized data.

### 3.2. Training Process

The input data window used in the proposed model to predict the next hour traffic $x_t$ is 3 h data before, i.e., $x_{t-1}$, $x_{t-2}$, and $x_{t-3}$. We set the historical data from 80 to 120 h to predict internet traffic activity from 10 to 25 h. In performing the training stage, we use the 90% of the dataset for the training step and the remaining data for the testing step because the size of the dataset is not too big. For the validation set, we select 10% of the training dataset. We set the epochs by 1000 and use mean square error (MSE) as the loss function in training process. We also present the loss instance of training and validation of each epoch to reflect on the performance of our proposed model during the training process as shown in Figure 5. The learning curve of the training and validation loss is high at the beginning. Then, it gradually decreases upon adding training examples, and it flattens. This means the proposed model is not overfitting.

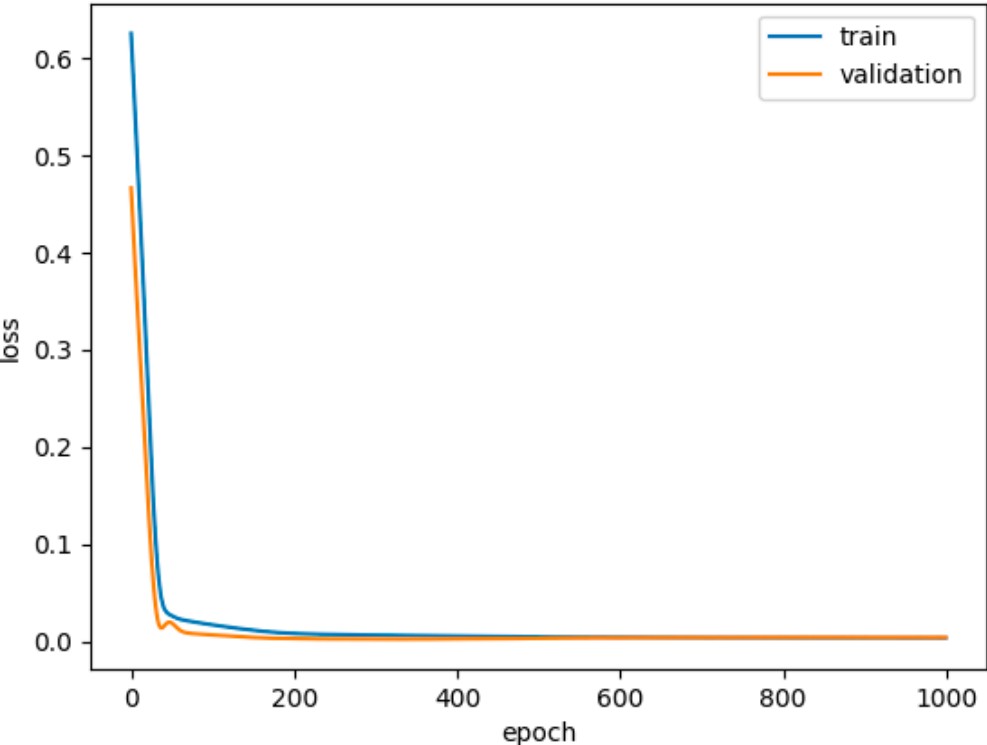

**Figure 5.** Training and validation losses.

### 3.3. Evaluation Metrics

To evaluate and know the quality of the proposed model, we present mean absolute percentage error (*MAPE*), root mean square error (*RMSE*), and mean absolute error (*MAE*) as evaluation metrics in the predicted data and the original data during the testing process as follows [21]:

$$MAPE = \frac{100\%}{T} \sum_{t=1}^{T} \left| \frac{v_{pd} - v_{ad}}{v_{ad}} \right|, \tag{10}$$

$$RMSE = \sqrt{\frac{1}{T} \sum_{t=1}^{T} (v_{pd} - v_{ad})^2}, \tag{11}$$

$$MAE = \frac{1}{T} \sum_{t=1}^{T} \left| v_{pd} - v_{ad} \right|, \tag{12}$$

where $v_{ad}$ and $v_{pd}$ denote the actual data and the predicted data, while $T$ is the amount of data.

### 3.4. Results and Analysis

To verify the proposed model, we present other benchmark models as a comparison to our base station traffic prediction model. Here, the experimental stage is performed in the internet traffic of three base stations. The compared benchmark models are ARIMA, LSTM, LSTM–1DCNN, Wavelet-LSTM, and Standard STL-GRU. We present ARIMA as the traditional model and LSTM as the single model in time series analysis to be compared. LSTM–1DCNN as the hybrid model initially takes the input maps from the dataset to the LSTM cell, then the outputs are utilized as input to the 1DCNN model. Then, Wavelet-LSTM and standard STL-GRU models have a similar process to the model proposed. That is, these models initially decompose the time series signal into several components then are used as input data.

Furthermore, the prediction result of base station traffic is evaluated using MAPE, RMSE, and MAPE. We perform the testing stages five times for each cell and calculate the

average of the results to represent the prediction result of the models in each cell. The result of traffic prediction of the proposed model and benchmark models in three cells is shown in Tables 1–3.

**Table 1.** The comparison results of traffic prediction in cell 3350.

| Model | MAPE | RMSE | MAE |
|---|---|---|---|
| ARIMA | 18.96% | 0.026 | 0.023 |
| LSTM | 15.04% | 0.031 | 0.031 |
| LSTM–1DCNN | 11.53% | 0.025 | 0.021 |
| Wavelet-LSTM | 15.52% | 0.030 | 0.026 |
| Standard STL-GRU | 15.43% | 0.029 | 0.025 |
| The Proposed Model | 12.10% | 0.024 | 0.021 |

**Table 2.** The comparison results of traffic prediction in cell 5060.

| Model | MAPE | RMSE | MAE |
|---|---|---|---|
| ARIMA | 25.59% | 0.899 | 0.755 |
| LSTM | 11.73% | 0.589 | 0.480 |
| LSTM–1DCNN | 17.28% | 0.800 | 0.784 |
| Wavelet-LSTM | 16.77% | 0.744 | 0.625 |
| Standard STL-GRU | 16.80% | 0.736 | 0.632 |
| The Proposed Model | 11.02% | 0.564 | 0.455 |

**Table 3.** The comparison results of traffic prediction in cell 5864.

| Model | MAPE | RMSE | MAE |
|---|---|---|---|
| ARIMA | 14.50% | 0.200 | 0.160 |
| LSTM | 13.94% | 0.239 | 0.200 |
| LSTM–1DCNN | 13.86% | 0.237 | 0.199 |
| Wavelet-LSTM | 27.03% | 0.450 | 0.402 |
| Standard STL-GRU | 15.32% | 0.261 | 0.214 |
| The Proposed Model | 17.62% | 0.277 | 0.230 |

Based on the tables above, the proposed model obtains the optimal result. The proposed model result in the base station of cell 3350 is an excellent model with the best result and the lowest values of RMSE of 0.024 and MAE of 0.021. Then, the proposed model also obtains the optimal result with the lowest values of MAPE of 11.02%, RMSE of 0.564, and MAE of 0.455 in the base station of cell 5060. Then, the proposed model cannot outperform the ARIMA model in the base station of cell 5864, obtaining the lowest values in RMSE and MAE. However, the proposed model overall achieves outstanding results in the experimental results. The decomposition using RobustSTL outperforms the wavelet and standard STL decomposition in extracting the time series components, especially in the base stations of cells 3350 and 5060.

The decomposition of time series data proves better achieving the accurate prediction, especially using RobustSTL. This decomposition method in our proposed model can extract dynamic patterns of traffic data despite containing some burstiness and outliers. Figure 6 shows that the predicted data of the proposed model in the base station of cell 5060, as an instance, matches the actual data with the low error. The actual data in the red line and the predicted data in green have similar patterns.

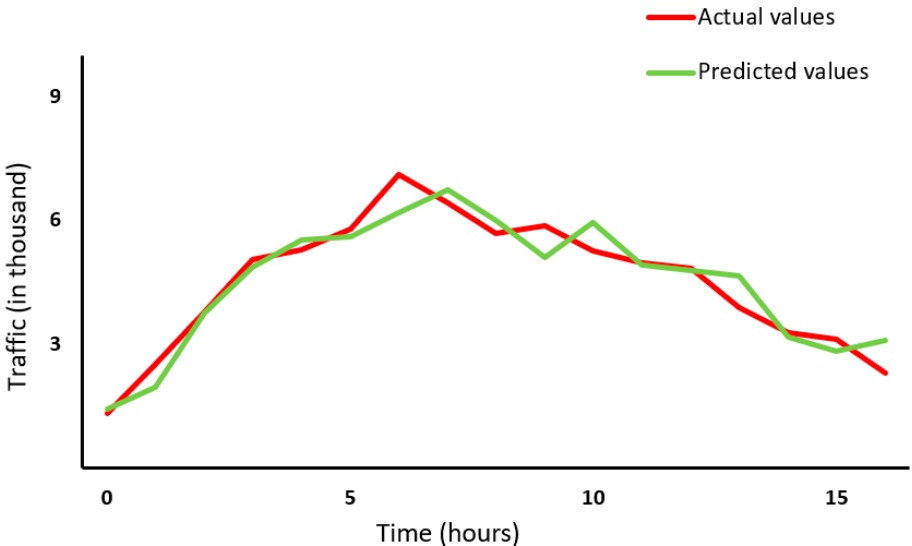

**Figure 6.** The comparison between the actual values and the predicted values.

## 4. Conclusions

Base station traffic prediction plays a vital role in green networking architecture in describing the future demands as references to apply sleeping control. The main challenges in base station traffic prediction are the dynamic and complicated patterns. Besides, some burstiness and outliers appear in the traffic. The decomposition process can increase the understanding of traffic rules and relationships. The decomposition using RobustSTL indicates the optimal result compared to counterpart schemes in the time series extraction. The combination of RobustSTL and 1DCNN-GRU generally obtains the optimal accuracy with the lowest values of MAPE, RMSE, MAE metrics in base station traffic prediction outperforming the other models. The proposed model can detect noises and outliers of traffic. For the next study, we hope the proposed model can be applied to a greater dataset with more features and a long series to evaluate the proposed model more comprehensively in the experimental stage.

**Author Contributions:** Conceptualization, C.-H.L.; methodology, U.N.; software, U.N.; validation, C.-H.L.; formal analysis, U.N.; resources, C.-H.L.; writing—original draft preparation, U.N.; writing—review and editing, U.N. and C.-H.L.; visualization, U.N.; supervision, C.-H.L.; project administration, C.-H.L.; funding acquisition, C.-H.L. All authors have read and agreed to the published version of the manuscript.

**Funding:** This work was financially supported by the Ministry of Science and Technology, Taiwan, under MOST- 110-2218-E-006-014-MBK and MOST- 110-2221-E-992-083.

**Institutional Review Board Statement:** Not applicable.

**Informed Consent Statement:** Not applicable.

**Data Availability Statement:** The data presented in this study are openly available in Kaggle (https://www.kaggle.com/andrewfager/analysis-and-modeling-of-internet-usage/data, accessed on 15 November 2021).

**Conflicts of Interest:** The authors declare no conflict of interest.

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
