# Peer review of "RobustSTL and Machine-Learning Hybrid to Improve Time Series Prediction of Base Station Traffic"

_electronics, doi:10.3390/electronics11081223_

Round 1
Reviewer 1 Report
The authors present a hybrid approach for time-series prediction of base station traffic. The paper is interesting and well-written, however, I have some concerns.
- Table 1 should be renamed to Algorithm 1. However, is this algorithm the same as Algorithm 1 of reference [12]? If yes, I think it should be removed because in your manuscript there are very few details that confuse the reader; a simple reference to the original algorithm of [12] should be more than enough.
- Is it possible that in Equation 4 instead of bi you should write bk ? And also, you should mention that m is the number of inputs. Generally, it would be helpful if you check all the equations and make sure that all variables are clearly mentioned and explained in the text.
- In Figure 2, the block "1 -" is a little confusing. Maybe you should add a "+1" and an "x(-1)" block or something similar.
- There is an extra parenthesis in Equation 8.
- Line 197 needs some rephrasing. It mentions that "GRU is the enhancement of the LSTM model, which is simpler" whereas GRU is simpler and LSTM is the enhancement.
- I am a bit confused in Figure 3. The three outputs of the RobustSTL are somehow concatenated forming a multivariable signal and then fed to the CNN? And what about GRU units? Are there three of them? And how are they connected to one another? I think the authors should elaborate more on this in the text and also make the connections clear in the figure.
- I think the author should describe their dataset and any preprocessing or normalization steps. Furthermore, a new section describing the training process should be added. This section should include the training/validation/testing set split, the number of epochs, the loss function, any callbacks used during training, and metrics during training/validation/testing (for example error for training and validation sets for all epochs). This information is valuable in order to understand that the model is trained properly and there is no overfitting.
- I think a point that should definitely be highlighted by the authors is the prediction horizon. How many hours ahead can the model predict? Also how often does the model need to predict? And what is the length of the input window?
Author Response
Manuscript ID: applsci-1659316
Responses to Reviewer 1
(For your information, the following paragraphs labeled with \(C)" and \(R)" indicate a review comment and our response, respectively. The change of text which has also been reflected in this manuscript is highlighted here in a gray box for better identification.)
- (C)
Table 1 should be renamed to Algorithm 1. However, is this algorithm the same as Algorithm 1 of reference [12]? If yes, I think it should be removed because in your manuscript there are very few details that confuse the reader; a simple reference to the original algorithm of [12] should be more than enough.
(R)
We are grateful for your comments. We already renamed Table 1 to Algorithm 1. This algorithm is the same as Algorithm 1 of reference. We still keep the algorithm to give the information process to readers, but we remove the details both in the algorithm and in section 2.3 to avoid confusion. We have revised the paragraph as following:
In standard STL, we assume that all components contained in time series data be-sides trend and seasonality are residual components. In contrast, RobustSTL proposed by Wen et al. [10] and adopted in this study claims that residual components can be further extracted into two terms, i.e., spike and noise. Moreover, RobustSTL can accurately and precisely decompose the time series data despite containing a long seasonality and high noise [10]. This method can handle fractional and shifted seasonality components over a period.
Data collected from the management unit frequently contains various noises. To get the exact trend and seasonality components, such noises initially need to be removed. Here, RobustSTL proposes a bilateral filtering to remove various noises. Then, RobustSTL utilizes LAD (least absolute deviations) with L1 norm regularizations to extract the trend component and non-local seasonal filtering to obtain the seasonality component. Given a data of time series, Algorithm 1 shows the decomposition summary of the RobustSTL method:
Algorithm 1. RobustSTL decomposition summary |
Input: , parameter configuration |
Output: |
1: Denoise the network traffic data by bilateral filtering to get denoised data |
2: Get the relative trend and apply this equation to denoised data |
3: Perform the seasonality extraction to using non-local seasonal filtering to get |
value |
4: Obtain trend, seasonality, and residual components |
, , |
5: repeat steps 1-4 to obtain more accurate estimation |
- (C)
Is it possible that in Equation 4 instead of bi you should write bk And also, you should mention that m is the number of inputs. Generally, it would be helpful if you check all the equations and make sure that all variables are clearly mentioned and explained in the text.
(R)
We appreciate your suggestion. We have revised Equation 4 in our manuscript to rewrite bi to bk and mention that m is the number of inputs. We also have checked all equations to make sure that each symbol can be understood.
- (C)
In Figure 2, the block "1 -" is a little confusing. Maybe you should add a "+1" and an "x(-1)" block or something similar.
(R)
Thanks for your correction. We have seen again the figure to redraw and redefine the block "1 -" as shown in Figure 3. What we mean by the block is "1 - Zt".
- (C)
There is an extra parenthesis in Equation 8.
(R)
We are very grateful again for your correction. We have checked and rewritten this equation to remove the extra parentheses in Equation 8. We have revised the following equation.
. |
(8) |
- (C)
Line 197 needs some rephrasing. It mentions that "GRU is the enhancement of the LSTM model, which is simpler" whereas GRU is simpler and LSTM is the enhancement.
(R)
Thanks, we mean that GRU is simpler than LSTM. Then, we already rephrased this sentence in first sentence of section 2.5 as following:
The GRU architecture is simpler model compared to the LSTM architecture, because the GRU is without memory cells.
- (C)
I am a bit confused in Figure 3. The three outputs of the RobustSTL are somehow concatenated forming a multivariable signal and then fed to the CNN? And what about GRU units? Are there three of them? And how are they connected to one another? I think the authors should elaborate more on this in the text and also make the connections clear in the figure.
(R)
This is a well-taken comment. We have re-explained this clearly in section 2.1. A raw input is decomposed by RobustSTL to extract trend, seasonality, and residual components. These separately are fed into 1DCNN as input. Then, these inputs are fed to the convolutional layer having 32 filters. The outputs of the convolutional layer are carried to the pooling layer. The fully connected layer receives the input from the pooling layer. The outputs from the 1DCNN block are put into the GRU block. The number of GRU cells is the same as the number of 1DCNN outputs. Then, the outputs of GRU are flattened with the fully connected layer to get the final output. We also redraw the figure to more be understood as shown in Figure 2 or line 120.
As we mentioned above, we use the combination model of RobustSTL and 1DCNN-GRU to predict base station traffic. Figure 1 shows that a raw input is decomposed by RobustSTL to reveal the underlying insights of base station traffic. After getting three decomposed components, i.e., the trend, seasonality, and residual components, these are simultaneously fed into 1DCNN. 1DCNN can help a better understanding of traffic patterns and spatial features. The 1DCNN architecture here consists of convolution, pooling, and fully connected layers. The number of inputs in the 1DCNN model from one raw input is 3 (three) obtained from the decomposition result. Then, these 1DCNN inputs are fed to the convolutional layer which has 32 filters. The outputs of the convolutional layer are carried to the pooling layer to minimize the dimension. The last layer in the 1DCNN is the fully connected layer where the input is from the pooling layer.
The outputs from the 1DCNN block are put into the GRU block. Through the GRU operation, the traffic characteristic and rules can be learned to improve the accuracy of the prediction result. The number of GRU cells is the same as the number of 1DCNN outputs. Then, the outputs of GRU are flattened with the fully connected layer to get the final output.
- (C)
I think the author should describe their dataset and any preprocessing or normalization steps. Furthermore, a new section describing the training process should be added. This section should include the training/validation/testing set split, the number of epochs, the loss function, any callbacks used during training, and metrics during training/validation/testing (for example error for training and validation sets for all epochs). This information is valuable in order to understand that the model is trained properly and there is no overfitting.
(R)
Thank for the valuable suggestion. We describe the dataset in section 3.1. Then, we create a new section describing the training process in section 3.2. We use 90% of data for the training set and the remaining for the testing set, because the size of the dataset is not too big. We take 10% of the training set for validation. We set the epochs by 1000 and use mean square error (MSE) as the loss function. We also present the instance of the training and validation loss curve for each epoch in Figure 5 showing no overfitting.
In performing training stage, we use the 90% of the dataset for training step and the remaining data for testing step, because the size of the dataset is not too big. For the validation set, we select 10% of the training dataset. We set the epochs by 1000 and use mean square error (MSE) as the loss function in training process. We also present the loss instance of training and validation of each epoch to reflect on the performance of our proposed model during training process as shown in Figure 5. Learning curve of the training and validation loss is high at the beginning. Then, it gradually decreases upon adding training examples and flattens. That means, the proposed model is not overfitting.
- (C)
I think a point that should definitely be highlighted by the authors is the prediction horizon. How many hours ahead can the model predict? Also how often does the model need to predict? And what is the length of the input window?.
(R)
Lastly, we say thank you for the comments. In our study, the proposed model can predict 10 to 25 hours ahead. We set the historical data from 80 to 120 hours as the historical data for the prediction reference to predict internet traffic activity. We set the length of the input window to predict the next hour traffic is 3 hours data before. We add this new explanation in first paragraph of section 3.2.
The input data window used in the proposed model to predict the next hour traffic xt is 3 hours data before, i.e., xt-1, xt-2, and xt-3. We set the historical data from 80 to 120 hours to predict internet traffic activity from 10 to 25 hours.

Reviewer 2 Report
- Do not include unnecessary paragraphs, which is not related with the topic like the first paragraph of the introduction section.
- The part 3 and 4 of the contribution paragraph are not the contribution of your paper. They are the result of the model.
- The RobustSTL and 1DCNN-GRU is originated from reference (12, 13, 14 and 15). What is difference between your idea and the references of [12, 13, 14 and 15]? Did you propose 1DCNN and GRU first time or proposed by other persons like [13, 14 or 15]? Specify them clearly in the paragraph between line 85-89.
- What is LAD stated at the line 138 of the page 3?
- The section 2.5 is the main idea. It should be described at the first in the section 2. This is a conceptual idea.
- The section 2.6 is used for evaluating the performance. So, it should be to the section 3.
- How many times did you test for each cell? It is the best value or a mean value? Describe them.
Author Response
Manuscript ID: applsci-1659316
Responses to Reviewer 2
(For your information, the following paragraphs labeled with \(C)" and \(R)" indicate a review comment and our response, respectively. The change of text which has also been reflected in this manuscript is highlighted here in a gray box for better identification.)
- (C)
Do not include unnecessary paragraphs, which is not related with the topic like the first paragraph of the introduction section.
(R)
We are grateful for your suggestion. We have already checked all paragraphs which do not relate to the topic and removed the first paragraph of the introduction section.
- (C)
The part 3 and 4 of the contribution paragraph are not the contribution of your paper. They are the result of the model.
(R)
Thank for your suggestion. We have seen again the points of our contribution. We remove parts 3 and 4. Then, we add a new part as a contribution point as following:
- The proposed model can give the reference for sleeping control operation in the base station by estimating the future internet traffic.
- (C)
The RobustSTL and 1DCNN-GRU is originated from reference (12, 13, 14 and 15). What is difference between your idea and the references of [12, 13, 14 and 15]? Did you propose 1DCNN and GRU first time or proposed by other persons like [13, 14 or 15]? Specify them clearly in the paragraph between line 85-89.
(R)
The difference between our idea and those references is that they did not design the hybrid model prediction of decomposition method and prediction means of deep learning. Originally, [13] proposed 1DCNN and GRU in the field of electricity consumption forecasting. However, it does not propose any decomposition methods. Then, we introduce RobustSTL in the beginning process to enhance the reliability of the prediction model. We revised this paragraph below in lines 70 to 85.
Although some models have reached well prediction of the network traffic with dynamic traffic patterns, they still need to improve in identifying abrupt changes, traffic burstiness, and outliers in future demands. Furthermore, this study proposes deep learning of base station traffic prediction with the reliability of traffic burstiness and outliers. To get the comprehensive pattern of traffic load from the base station, this study first decomposes the original traffic data using RobustSTL method, instead of the standard STL. RobustSTL not only decomposes seasonal, trend, and remainder or residual components but also further can decompose the remainder component into a spike and white noise [10]. Resulted components are synchronously fed into a hybrid model of one-dimensional convolutional neural network (1DCNN) and gated recurrent units (GRU). Through this hybrid model, 1DCNN can catch and extract features of the components [11] and GRU can capture the rules and relationships among components to improve the model performance [12]. Although there is a study in the load forecasting field proposing 1DCNN-GRU [13], it does not utilizes decomposition. Furthermore, we enhance their proposed model that not only proposes the hybrid model of 1DCNN and GRU but also utilize the decomposition of RobustSTL in the beginning stage.
- (C)
What is LAD stated at the line 138 of the page 3.
(R)
Thank you for the comment. Here, LAD is least absolute deviations. We have added the acronym of LAD in line 145.
Then, RobustSTL utilizes LAD (least absolute deviations) with L1 norm regularizations to extract the trend component and non-local seasonal filtering to obtain the seasonality component.
- (C)
The section 2.5 is the main idea. It should be described at the first in the section 2. This is a conceptual idea.
(R)
Thank you for your suggestion, we already moved the section 2.5 to the section 2.1 as shown in line 102.
- (C)
The section 2.6 is used for evaluating the performance. So, it should be to the section 3.
(R)
Thank you for your suggestion, we moved the section 2.6 to the section 3.3 as shown in line 247.
- (C)
How many times did you test for each cell? It is the best value or a mean value? Describe them.
(R)
We undertook the testing experiment of all prediction models for each cell five times. Then, we take the best prediction result of each model to represent their performance. We add this explanation in line 265 in second paragraph of the section 3.4 as following.
Furthermore, the prediction result of base station traffic is evaluated using MAPE, RMSE, and MAPE. We perform five times of testing stages for each cell and select their best prediction to represent the prediction result of the models in each cell. The result of traffic prediction of the proposed model and benchmark models in three cells is shown in Tables 2, 3, and 4.

Round 2
Reviewer 1 Report
The authors have addressed all of my concerns.
Author Response
There are no suggestions and comments from the first reviewer, so we do not need to give a response

Reviewer 2 Report
- You compared yoor system with other similar methods which use all LSTM (GRU is a LSTM). I think that you should compare your model with other method (traditional or a algorithmic approach).
- 2. In table 5 and 6, you showed the best results of test. However, it is desirable to show the average result.
Author Response
(For your information, the following paragraphs labeled with \(C)" and \(R)" indicate a review comment and our response, respectively. The change of text which has also been reflected in this manuscript is highlighted here in italics for better identification.)
- (C)
You compared your system with other similar methods which use all LSTM (GRU is a LSTM). I think that you should compare your model with other method (traditional or a algorithmic approach).
(R)
We are grateful for your suggestion. We have already added the ARIMA model as the compared model of the traditional approach. The acronym of ARIMA is mentioned in line 39 in the second paragraph of the introduction. Then, this revision can be seen in section 3.4 and Tables 1, 2, and 3.
The compared benchmark models are ARIMA, LSTM, LSTM-1DCNN, Wavelet-LSTM, and Standard STL-GRU. We present ARIMA as the traditional model and LSTM as the single model in time series analysis to be compared.
- (C)
In table 5 and 6, you showed the best results of test. However, it is desirable to show the average result.
(R)
Thank you for your suggestion. Maybe you mean Tables 1, 2, and 3. Based on your suggestions, we have changed the values in the table to show the experimental results are the average values of each model in the experiment in each cell. Furthermore, the revised manuscript can be seen in the second paragraph of section 3.4 and Tables 1, 2, and 3.
We perform five times of testing stages for each cell and calculate the average of the results to represent the prediction result of the models in each cell.
